# Airway Management and General Anesthesia in Pediatric Patients with Special Needs Undergoing Dental Surgery: A Retrospective Study

**DOI:** 10.3390/reports7030079

**Published:** 2024-09-17

**Authors:** Alessandra Ciccozzi, Ettore Lupi, Stefano Necozione, Filippo Giovannetti, Antonio Oliva, Roberta Ciuffini, Chiara Angeletti, Franco Marinangeli, Alba Piroli

**Affiliations:** 1Department of Life, Health and Environmental Sciences (MeSVA), University of L’Aquila, 67100 L’Aquila, Italy; alessandra.ciccozzi@univaq.it (A.C.); stefano.necozione@univaq.it (S.N.); filippo.giovannetti@univaq.it (F.G.); roberta.ciuffini@univaq.it (R.C.); franco.marinangeli@univaq.it (F.M.); 2San Salvatore Teaching Hospital of L’Aquila, 67100 L’Aquila, Italy; ett.lupi@gmail.com (E.L.);; 3Department of Anaesthesia, Mazzini Hospital, 64100 Teramo, Italy

**Keywords:** difficult airway, pediatrics, general anesthesia, psychophysical disorders

## Abstract

Background: The definition of patients with special needs (SNs) is used in the literature to refer to individuals with mental and physical disorders for whom the usual perioperative pathways are not applicable due to lack of cooperation, regardless of age. Studies in the literature recognize the appropriateness of general anesthesia for performing day surgery dental care in this type of patient. Objectives: The main objective was to assess the possible incidence of difficult airway management, understood as difficulty ventilating and/or intubating the patient. A secondary objective was to highlight the influence of general anesthesia on patient outcomes by testing the incidence of perioperative complications. Methods: The present retrospective, single-center, observational study involved 41 uncooperative patients aged between 3 and 17 undergoing dental surgery under general anesthesia. Data relating to airway management and general anesthesia present in the medical records were analyzed. Results: Tracheal intubation was successfully completed in all of the patients considered, and in no case did the patient have to be woken up because of difficulty in airway management. No perioperative complications attributable to anesthesia were found in any patients. Conclusions: From the present experience, it can be concluded that general anesthesia is a suitable option for performing dental care in pediatric subjects with special needs, and that although the peculiar perioperative management of these patients might increase the risk of possible anesthesia-related side effects, no complications have been encountered in any case.

## 1. Introduction

Pre-, intra-, and postoperative management of patients with special needs (SNs) to undergo dental treatment must be scrupulously planned in order to make the perioperative procedure safe, comfortable, tailored to the specific needs of these patients, and as minimally traumatic as possible [1].

The term SNs is used in the literature to define patients with psychophysical, relational, and cognitive disorders, for whom the usual perioperative course of action is inapplicable due to several critical issues and for whom both surgical and especially anesthesiological procedures are burdened with a sometimes greatly increased risk, whether due to the patient’s morphological–structural characteristics, the underlying disease, or often the coexistence of complex comorbidities [2].

Patients with SNs are at increased risk of oral disease throughout their lives [1]. Several factors explain this high incidence: poor hygiene, psychotropic medications, type of diet, lack of access to dental care, and difficulty with dental assessment [3].

These patients, moreover, due to the difficulty of history taking, suitable anesthesiological risk assessment, and lack of cooperation, frequently fail to meet the outpatient management criteria applied to other patients [4].

In addition to these difficulties, SNs patients may also have an increased risk of difficult airway management due to the frequent presence of anatomo-functional malformations of the facial massif (Down’s syndrome and other genetic syndromes) and/or thorax (infantile cerebral palsy), gastroesophageal reflux from impaired functioning of the upper esophageal sphincter (infantile cerebral palsy), and obesity (Down’s syndrome), with a concomitant higher incidence of postoperative complications [5].

Studies in the literature recognize the appropriateness of general anesthesia for day surgery dental care in uncooperative patients or those with simple phobia [6,7].

In fact, sedation alone, although an effective and safe alternative to general anesthesia, is frequently unsuccessful in patients with severe cooperation problems, both in terms of maintaining airway patency and in relation to the duration and type of dental procedure. Because of the latter issues, general anesthesia appears to be indicated in uncooperative patients with SNs to perform dental treatment.

Few studies performed on SNs pediatric patients undergoing dental treatment under general anesthesia have evaluated the incidence of difficult airway management and any perioperative complications.

The main objective of the present observational, retrospective, single-center study was to evaluate the incidence of difficult airway management (difficulty with ventilation and/or intubation) in pediatric uncooperative SNs patients, with psychophysical and/or cognitive disorders, undergoing dental treatment under general anesthesia.

The secondary objective was to verify the type of general anesthesia performed and highlight its influence on the patients’ outcomes, looking for the possible occurrence of major (cardiovascular, respiratory, neurological) and minor (nausea, vomiting, shivering, pain) perioperative complications and evaluating the patients’ awakening time from anesthesia and hospital stay.

## 2. Materials and Methods

For the execution of this retrospective, observational, single-center study, the records of uncooperative patients admitted to the Department of Maxillofacial Surgery at P.O. San Salvatore in L’Aquila during the period January 2021 to August 2022 were examined. All the collected data were used with the specific authorization of the Presidio Medical Directorate of the S. Salvatore Civil Hospital in L’Aquila and after obtaining approval to carry out the study from the Ethics Committee for the Provinces of L’Aquila and Teramo (minute No. 25 of 14 September 2022).

After verifying the presence in the medical record of consent to the recording and use, for educational and/or scientific purposes, of data or other clinical documentation acquired during the course of treatment (Art. 10, Law No. 675 of 31 December 1996 on the protection of persons with respect to the processing of personal data), signed by the parent/guardian at the same time as the consent to the performance of general anesthesia, all uncooperative patients aged between 3 and 17 years old, with ASA status I-III, undergoing dental surgery under general anesthesia during the period under consideration were included in this study.

From the analysis of medical records, the following information was extracted and recorded on a special form structured into six main areas:Patient-reported data: age, sex, weight, BMI, ASA Status, main disability type, comorbidities, and current therapies;Predictive data of possible increased anesthesiologic risk: presence of craniofacial and early airway anatomical malformations, gastroesophageal reflux, myopathies, epilepsy, severe deformities of the cervical and/or thoracic region; presence/absence of predictive scores of difficult intubation, physical examination, blood chemistry or instrumental tests;Data related to the surgical procedure: type of dental procedure performed, duration of surgery, any secondary, diagnostic or surgical procedures associated with the main one and performed while the patient is anesthetized (blood chemistry tests, plasma dosages of psychotropic drugs, electrocardiogram, electroencephalogram, ultrasound study of organs, endoscopic examinations, gastrointestinal examinations, minor surgery, specialist evaluations);Anesthesia data:
Type, dosage, and timing of administration of anesthetic premedication;Score of sedation on arrival in the operating room (0 = awake and agitated [insufficient]; 1 = awake and calm [sufficient]; 2 = sleepy but can be awakened [adequate]; 3 = sleepy and difficult to awaken [excessive]);Pre-induction venous cannulation;Type of anesthesia induction: inhalation or intravenous;Drugs used to maintain general anesthesia;Intraoperative monitoring employed: cardiorespiratory (ECG, pulse oximetry, capnography) and anesthesiologic specialist (Bispectral Index [BIS] and curarization [TOF] analysis)Wake-up time (from the end of surgery to extubation).Airway management data:
Possible difficulty with ventilation in face mask;Use of the laryngeal mask;Monitoring device chosen for intubation;Intubation mode: orotracheal or nasotracheal;Time required for endotracheal intubation, defined as the time between the start of the first laryngoscopy and visualization of the third capnographic curve;Possible difficulty with intubation, defined as the need to perform more than 1 attempt, need to change devices, need to change operators, or need to awaken the patient;Any complications that occurred during the procedure.Perioperative data: Postoperative analgesic therapy; length of hospital stay; possible occurrence of major (cardiovascular, respiratory, neurological) or minor (nausea vomiting, chills, pain) complications.

For the protection of patients, all information was recorded anonymously, and the data collection form did not provide for the recording of information that would allow the identification of the patient. For patients who had undergone more than one dental procedure during the period under consideration, only one form was filled out, recording only data on the first procedure performed in chronological order.

Statistical analysis was conducted using “The Sas System version 9.4”. Descriptive analysis was carried out using the mean, standard deviation, and range for continuous quantitative variables and with frequencies for nonquantitative variables. After assessing the normality of the distribution of the variables with the Shapiro–Wilk test, statistical comparisons were carried out using the non-parametric Wilcoxon Rank Sum test.

## 3. Results

At the Department of Maxillofacial Surgery at San Salvatore Hospital in L’Aquila, 56 uncooperative patients, aged between 3 and 17 years old, who were candidates for dental surgery under general anesthesia, were admitted during the period from 1 January 2021 to 31 August 2022. The procedure was performed in 54 patients, while in two cases the procedure could not be performed due to health issues detected at the time of admission, which made it necessary to postpone the procedure. Thirteen cases were excluded because they referred to dental procedures performed on patients previously treated during the reporting period. A total of 41 cases were deemed useful for the investigation. Table 1 describes the composition of the patient sample according to demographic and clinical characteristics, including the main disability (disorder reported in the medical record as main) and comorbidities presented (disorders coexisting with the main pathology).

As can be seen in the table, 25 patients had autism spectrum disorder as their main disability, which was therefore the most represented condition (60.98%), followed, albeit with a wide gap, by spastic tetraparesis and/or infantile cerebral palsy and Down’s syndrome, present in 3 patients, respectively.

Among the comorbidities presented, understood as conditions associated with the main disability and potentially predictive of increased difficulty in anesthesiological management, epilepsy was found most frequently, present in 73.17% of the sample considered, although it only represented the main pathology in two patients.

In accordance with these data, antiepileptic drugs were the most widely used, resulting in the home therapy of 78.04% of patients, followed by antipsychotic drugs taken by 26.82% of patients (Table 2).

Regarding the type of surgery performed, oral cavity check/clearance was the dental procedure performed in all patients: in 19 cases as the only treatment, in 15 patients combined with dental extractions, in 3 patients combined with dental extractions and caries treatment, in 2 patients combined with excision of a facial and/or tongue neoformation, and in 2 patients combined with dental impression (Table 3).

Furthermore, in four patients, in addition to the planned dental surgery, other medical procedures were performed, taking advantage of the general anesthesia performed: a urological examination with replacement of the suprapubic catheter, a gastroenterological examination with esophagogastroduodenoscopy (EGDS), an ENT examination, and a percutaneous endoscopic gastrostomy (PEG) replacement.

During preoperative anesthesiological evaluation, predictive indices of difficult airway management could not be assessed in 26 patients (63.41%), routine preoperative hematochemical examinations could not be performed in 17 patients (41.46%), an ECG could not be performed in 7 patients (17.07%), and objective examination could not be performed in 2 patients (4.88%). In these patients, electrocardiographic recording and blood sampling were performed with the patient anesthetized, immediately before surgery.

Anesthetic premedication, given in all cases with midazolam per os, was administered in the inpatient ward to all patients, and upon arrival in the operating room, sedation was rated adequate in 58.54% of patients and sufficient in 7.32% of cases, while as many as 11 patients arrived in the operating room frankly agitated (Table 4).

In 12 patients, cannulation of a vein before induction of anesthesia was not possible, and thus was induced by employing inhaled drugs (steal inhalation induction), followed by intravenous drugs as soon as possible. Table 5 shows the drugs used for induction and maintenance of general anesthesia.

Intraoperative monitoring of vital parameters included continuous recording of electrocardiographic (ECG) tracing, Heart Rate (HR), Pulse Oximetry (SaO_2_), and end-expiratory CO_2_ (EtCO_2_) in all cases and neuromuscular transmission by TOF (Train of Four) in eight cases, while depth of anesthesia was only monitored by Bispectral Index (BIS) in two patients.

For all subjects in the sample taken for analysis, there was no difficulty with face mask ventilation after induction of anesthesia, and no need for laryngeal mask placement.

Endotracheal intubation was successfully completed in all patients, and in no case were complications reported or did the patient have to be awakened due to difficulty in airway management. Table 6 shows the main data regarding endotracheal intubation.

In 37 patients, intubation was performed with a single attempt, in 3 patients 2 attempts were made, and in only 1 case 3 attempts were needed.

In four patients, intubation difficulty was recorded in the chart: In one case, there was a planned difficult orotracheal intubation initially attempted with the video-laryngoscope (Glidescope^®^, Bothell, DC, USA); after failing two attempts, intubation was completed by the same operator using the direct laryngoscope (Macintosh^®^, Bothell, DC, USA); in the remaining three cases, however, orotracheal intubation was performed using the video-laryngoscope (Glidescope^®^) in one patient as the first and only monitoring device and in the remaining two as the second device after failure of intubation with direct laryngoscopy. Only in one case was it necessary to change operators.

No statistically significant differences were found, using Wilcoxon’s test, between intubation difficulty and patients’ age (*p* = 0.0572) or BMI (*p* = 0.9793). It should be noted, among other things, that only one patient was obese.

The mean time to awakening from anesthesia was 14.02 ± 4.99 min (range = 5–30 min), and the operative and inpatient times are shown in Table 7.

Postoperative analgesic therapy was given in 92.68% of patients, and the most commonly used drug was paracetamol, administered as the sole analgesic drug in 26 patients; paracetamol was also used in combination with NSAIDs in six patients and in combination with morphine in three (Table 8).

Finally, from the analysis of the data recorded in the charts, it is reported that there were no major perioperative complications referable to anesthesia.

## 4. Discussion

For uncooperative pediatric or adult patients, it is necessary to use general anesthesia for dental treatments that in the general population are usually performed with outpatient procedures, without the need for sedation or anesthesia [8]. General anesthesia is the first option chosen by health care providers because it avoids undue stress on the patient while providing high-quality dental treatment due to its guaranteed safety and efficacy, despite the potential risks associated with it [9,10].

Given lack of cooperation and the difficulty in communicating with a patient with special needs, regardless of age, the anesthesiologist often has to rely on the parent or caregiver to collect the pathological and pharmacological history. The latter is important because the uncooperative patient takes numerous medications at home, even in pediatric age, which could interfere with the medications used for premedication, anesthesia, and pain management [11].

In special needs patients, moreover, preoperative assessment of predictive indices of difficult airway management is often not possible, and it is not always possible to perform the necessary laboratory tests or instrumental examinations, such as routine blood chemistry tests and electrocardiogram, normally performed in patients who are candidates for anesthesia [4].

In addition, a noncooperative pediatric patient may present with peculiarities directly related to the underlying pathology: for those with autism spectrum disorder, difficulties in cooperation and communication and possible drug interactions between home therapy and drugs administered for anesthetic premedication and anesthesia [12,13]; for those with Down’s syndrome, the presence of congenital heart defects, peculiar craniofacial anatomical features, atlantoaxial instability, and obesity [13,14,15]; for those with infantile cerebral palsy or spastic tetraparesis, sialorrhea, scoliosis, gastroesophageal reflux, and low subcutaneous fat [16,17]; finally, for those with epilepsy, there may be drug interactions between antiepileptic therapy and drugs administered as anesthetic premedication or as anesthetics, with an increased risk of seizures [18].

Particular anatomical features, medication taken at home, and any comorbidities present may contribute to difficult preoperative, intraoperative, and postoperative management of the patient, with an increased risk of difficult airway assistance and anesthesiologic complications. Difficulty in airway management of pediatric patients is not uncommon in anesthesiology practice and can be a cause of morbidity and mortality [19].

The basic principles for the safe management of difficult airways in the pediatric patient are essentially superimposed on those provided for adult patients: recognition, preparation, and planning of management strategies. However, some peculiarities should be emphasized in the pediatric setting, which are particularly important when the patient is also uncooperative. In particular, the physical assessment phase is often incomplete or even absent because the patient is not cooperative in performing the necessary maneuvers [20]. Syndromic or malformative clinical pictures, which may make it difficult to ensure adequate oxygenation and ventilation, should also be appropriately considered in the context of the most relevant conditions of difficult airway in pediatric age [21]. In addition, airway management in maxillofacial surgery may present foreseeable additional difficulties, due to the proximity between the tracheal tube and the surgery field [22]. 

This observational, retrospective, single-center study was primarily aimed at evaluating the incidence of difficulty in airway management, understood as difficulty with ventilation and/or difficulty with intubation, in uncooperative patients aged 3 to 17 years undergoing dental treatment under general anesthesia.

For all 41 patients examined, there was no difficulty with pre-intubation ventilation, which was carried out in all cases with a face mask, and in no patient was it necessary to use a laryngeal mask.

There were only four patients who presented intubation difficulties (9.75%), and in all cases the procedure was completed—in three cases using the video-laryngoscope and in only one patient by direct laryngoscopy. For the three patients intubated using the video-laryngoscope, it can be seen that in two cases the technique with direct laryngoscopy was used first, and following a failed attempt, the video-laryngoscope was promptly used successfully. This finding is in agreement with the most recent data in the literature, which highlight the important role of the video-laryngoscope in improving glottic vision and increasing the success rate of tracheal intubation [23,24,25].

It is more difficult to explain the dynamics in the one patient with intubation difficulties who was intubated by direct laryngoscopy after the failure of two attempts made with the video-laryngoscope. Analysis of the anesthesiology record does not clearly reveal the reasons for this failure; however, one can speculate on an important role played by both the patient’s peculiar anatomy and issues related to the use of the video-laryngoscope. In fact, this was an 11-year-old male patient diagnosed with severe epileptic encephalopathy and with a report in the anesthesiology chart of possible difficult airway management due to facial conformation and macroglossia, although the predictive scores for difficult intubation could not be detected. In this case, moreover, for reasons not described, the video-laryngoscope used to perform the procedure was not the Glidescope^®^, used in all other cases, but rather a handheld device (Tuoren^®^, Hangzhou, China), equipped with a smaller screen and certainly not routinely used by anesthesiologists. This may have compromised the success of the maneuver.

Importantly, no patients were awakened because of the difficulty in airway management, and the planned intervention could be performed in all cases. Although in 26 patients, i.e., 63.41% of the sample, predictive indices of difficult airway management could not be assessed, an assessment considered necessary to ensure patient safety in the Operating Room [26], a difficulty with intubation, in all cases promptly resolved, was described in only four patients (9.75%).

In 90.24% of the patients, intubation was performed on the first attempt, and the average time taken, reported in the chart as the time from the start of the first laryngoscopy to the display of the third capnographic curve on the monitor, was 3 min and 32 s.

We must also point out that during the preoperative visit, routine blood tests could not be performed in 41.46% of patients; ECGs could not be performed in 17.07% of patients and objective examination in 4.88% patients. All of this represents clinical information that is important for a safe preoperative course, the lack of which inevitably makes anesthesia more risky. In these patients, blood sampling and ECG were performed after the induction of anesthesia; this is especially useful to provide the patient with a clinical record that may be available in the event of a subsequent new surgical procedure, or to perform further instrumental investigations that may be necessary [2].

In 9.76% of patients, in addition, specialized procedures that the patient needed and had not been performed previously were performed under general anesthesia precisely because of the lack of cooperation.

No statistically significant differences were found between intubation difficulty and the patients’ age or BMI.

The secondary objective of the study was to highlight the influence of general anesthesia on patient outcomes, understood primarily in terms of perioperative complications, time to wake up from anesthesia, and hospital stay.

The time to wake up from general anesthesia in the studied sample averaged 14.02 ± 4.99 min (range = 5–30 min), with a median of 15 min; this finding shows that general anesthesia in these patients did not involve particularly prolonged awakening times.

The total length of hospital stay averaged 11 h and 31 min, with variability from a low of 3 h to a high of 144 h, found in only one patient and attributable to surgical complications that necessitated longer hospitalization. Considering these data, it is possible to say that although the dental procedures were performed under general anesthesia, almost all of the procedures were conducted under day surgery, with a short length of stay. This has significant benefits as it greatly reduces the stress of young patients and the discomfort of carers.

In no case were there any intra- or postoperative complications, minor or major, attributable to anesthetic conduct, despite the numerous medications taken by most of the patients considered, with potential interactions with the anesthetics used.

Despite these limitations, the present retrospective may suggest that general anesthesia is an excellent option for performing dental care in pediatric patients with special needs, and although the procedure is not without possible side effects, no complications have been encountered in any case.

For the administration of general anesthesia in uncooperative patients to be safe and ensure a good outcome, careful management of subjects during the preoperative, intraoperative, and postoperative periods is necessary, and adequate attention should be paid to airway management. Such precautions are particularly important precisely because, often, in such patients it is not possible to perform a comprehensive baseline history, assess predictive indices of difficult airway management, perform objective examination, perform laboratory and instrumental investigations, and it is not possible, in fact, to implement perioperative management equal to that offered to the general population. The increased attention that is required for patients with special needs ensures, however, reasonably safe anesthesia that allows the performance of surgeries and procedures that would otherwise not be performed in this type of patient. In conclusion, many pediatric patients with various disabilities require care that can only be performed under general anesthesia; therefore, sharing experiences on anesthesia and airway management in these patients is useful to improve clinical care and safety. Future studies are needed to better identify risk factors, evaluate treatment algorithms, and determine the best practice in the perioperative management of pediatric patients with special needs.

## 5. Study Limitations

The fact that this study was carried out as a retrospective evaluation can be considered a limitation, as this research mode allowed only the data recorded in the patients’ medical records to be analyzed and, therefore, did not make it possible to investigate other aspects that might be of interest, such as the satisfaction of family members regarding perioperative management, their suggestions, if any, or the presence of agitation of patients post-surgery, in the ward or at home. Another limitation of the study is the small sample size, mainly due to the COVID-19 health emergency present in Italy in the period under consideration, which numerically limited the execution of all elective interventions, including dental practices.

## Figures and Tables

**Table 1 reports-07-00079-t001:** Distribution of the patients according to demographic and clinical characteristics.

Total N = 41	No.	%	Range
Gender			
Males	24	58.53	
Females	17	41.46	
Age (years)			
$\bar{X}$ ± SD	11.22 ± 3.68		4–17
Weight (Kg)			
$\bar{X}$ ± SD	42.65 ± 19.48		15.50–92.00
Height (cm)			
$\bar{X}$ ± SD	144.63 ± 20.99		99.00–182.00
BMI	19.43 ± 4.22		12.80–30.40
ASA Status			
I	6	14.63	
II	29	70.73	
III	6	14.63	
Main Disability			
-Autism spectrum disorder	25	60.98	
-Spastic tetraparesis/	3	7.32	
Infantile cerebral palsy			
-Down’s syndrome	3	7.32	
-Epilepsy	2	4.88	
-Encephalopathy	2	4.88	
-Other	6	14.63	
Comorbidity			
Epilepsy	30	73.17	
Cervico-thoracic anomalies	7	17.07	
Muscular disorders	1	2.44	
Esophageal gastric reflux	1	2.44	
Airway anomalies	1	2.44	
Craniofacial anomalies	1	2.44	

**Table 2 reports-07-00079-t002:** Home therapy (* Drug taken by one patient only).

Drugs	No.	%
Antiepileptic	32	78.04
Antipsychotic	11	26.82
Anxiolytic	6	14.63
Myorelaxant	4	9.75
Antidepressant	3	7.31
Gastroprotector	3	7.31
Thyroid hormones	2	4.87
* Other	13	31.70

**Table 3 reports-07-00079-t003:** Main dental procedure.

Type of Main Procedure	No.	%
Oral cavity check/clearance	19	46.34
Dental extractions	15	36.59
Dental extractions + caries treatment	3	7.32
Excision of facial and/or tongue neoformation	2	4.87
Dental impression	2	4.87

**Table 4 reports-07-00079-t004:** Sedation scores.

Sedation Score on Arrival in the Operating Room	No.	%
0 = awake and agitated (insufficient)	11	26.83
1 = awake and calm (sufficient)	3	7.32
2 = sleepy but can be awakened (adequate)	24	58.54
3 = sleepy and difficult to awaken (excessive)	3	7.32

**Table 5 reports-07-00079-t005:** Drugs used for the induction and maintenance of anesthesia.

Induction Drugs	No.	%
Propofol + Rocuronium + Remifentanil	16	39.02
Sevoflurane + Propofol + Rocuronium + Fentanyl	11	26.83
Sevoflurane + Propofol + Rocuronium + Remifentanil	8	19.51
Propofol + Sevoflurane + Rocuronium	4	9.76
Ketamine	2	1.92
Maintenance drugs		
Sevoflurane + Remifentanil	27	65.85
Sevofluorane	6	14.63
Desflurane + Remifentanil	4	9.76
Propofol + Remifentanil	3	7.31
Ketamine	1	2.44

**Table 6 reports-07-00079-t006:** Summary data of endotracheal intubation.

	No.	%	Range
Technique employed			
Videolaryngoscopy (VDL)	33	80.49	
Direct laryngoscopy (DL)	5	12.19	
DL + VDL	3	7.32	
Access			
Orotracheal	33	92.45	
Nasotracheal	8	7.54	
Number of attempts			
1	37	90.24	
2	3	7.31	
3	1	2.44	
Intubation time (min)			
$\bar{X}$ ± SD	3.32		1.47–10.32
Need to change device			
Yes	3	7.50	
No	38	92.68	
Need to change operator			
Yes	1	2.44	
No	40	97.56	
Intubation difficulty			
Yes	4	9.75	
No	37	90.24	
Need to wake up the patient			
Yes	0	0%	
No	41	100%	

**Table 7 reports-07-00079-t007:** Intraoperative and hospital times.

	$\bar{X}$ ± SD	Range
Duration of intervention (min)	46.59 ± 23.73	10–130
Length of stay in the operating room (min)	104.57 ± 31.26	50–250
Time between discharge from operating room-discharge from hospital (h)	6.35 ± 12.99	1–113
Total length of hospital stay (h)	11.31 ± 16.28	3–144

**Table 8 reports-07-00079-t008:** Postoperative pain therapy.

Drugs Used	No.	%
Paracetamol	26	63.41
Paracetamol + NSAIDs	6	14.63
Paracetamol + Morphine	3	7.32
Morphine	3	7.32
None	3	7.32

## Data Availability

The original data presented in this study are available on reasonable request from the corresponding author. The data are not publicly available due to privacy.

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
