# Peer review of "Airway Management and General Anesthesia in Pediatric Patients with Special Needs Undergoing Dental Surgery: A Retrospective Study"

_reports, 2024, doi:10.3390/reports7030079_

Round 1

Reviewer 1 Report (Previous Reviewer 3)

Comments and Suggestions for Authors

The previous comments still stand. The description is not novel enough to warrant publication.

Author Response

Comments 1: The previous comments still stand. The description is not novel enough to warrant publication.

Response 1: Thank you very much for taking the time to review this manuscript. We believe that it is still useful to address the issue of care for patients with special needs. We hope that even a small single-center study reporting its favorable experience can help other healthcare professionals to better address the problem.

Reviewer 2 Report (New Reviewer)

Comments and Suggestions for Authors

Dear Authors,

Thank you for the opportunity of reviewing such an interesting manuscript that touches a very important problem. I realize how much work you put into the survey. However, in my opinion, this manuscript is not ready for publication because there are some serious issues that the Authors need to address before the manuscript can be considered for publication.

1. I tried to find the answer to "the secondary objective was to verify the type of general anesthesia performed and highlight its influence on the patients' outcome"- I could not find the connection between the type of anesthesia and the patients' outcome. Please explain and supply. 

2. Did you observe any postoperative adverse events (e.g. vomiting etc.)? Was it connected with lower sedation score (I mean agitation)?

Author Response

Thank you very much for taking the time to review this manuscript.

Comments1: I tried to find the answer to "the secondary objective was to verify the type of general anesthesia performed and highlight its influence on the patients' outcome"- I could not find the connection between the type of anesthesia and the patients' outcome. Please explain and supply. 

Response 1: Thank you very much for taking the time to review this manuscript. The data reported in this retrospective study are those present in the examined medical records. In particular, the drugs used for induction and maintenance of general anesthesia are reported in Table 5; the data relating to endotracheal intubation are reported in Table 6. From the examination of the medical records, no patient had complications attributable to general anesthesia. It was therefore not possible to highlight a connection between the type of general anesthesia performed and patients’ outcome.  With regard to the influence of general anesthesia on patient outcomes, understood primarily in terms of perioperative complications, time to wake up from anesthesia, and hospital stay, are reported on pages number 9 and 10; lines 322-338.The important message in our opinion is “the present experience may suggest that general anesthesia is an excellent option for performing dental care in pediatric patients with special needs, and although the procedure is not without possible side effects, no complications have been encountered in any case.” (page number 10; lines 339-342)

Comments 2: Did you observe any postoperative adverse events (e.g. vomiting etc.)? Was it connected with lower sedation score (I mean agitation)?

Response 2: From the analysis of the medical records in no case were there any minor or major postoperative adverse events. The sedation score was only related to the arrival in the operating room (verification of the effectiveness of the premedication) (page number 3, lines 140-142). The sedation (or agitation) score upon awakening was not present in the medical record. Therefore, the data were not reported.

Reviewer 3 Report (New Reviewer)

Comments and Suggestions for Authors

Tables need to be refferred in text as Table nr. 1, 2..

What was the main pathology in patients where there could not be blood harvesting before procedure?

Discussion section needs good improvement. There is a high lack of literature studies report either comparison.

Conclusion is vague.

References section needs to be revised.

Comments on the Quality of English Language

Grammar and spelling in text and in References also need revision.

Author Response

Thank you very much for taking the time to review this manuscript. Please find the detailed responses below and the corresponding revisions/corrections highlighted (in red) changes in the re-submitted files.

Comments 1: Tables need to be refferred in text as Table nr. 1, 2..

Response 1: Thank you for pointing this out.  All tables have been indicated in the text as Table nr. 1, 2 … (in red in the text ), as suggested.

Comments 2: What was the main pathology in patients where there could not be blood harvesting before procedure?

Response 2: The patients with “special needs” considered in this study are often completely uncooperative. This leads in many cases to the impossibility of carrying out procedures (blood sampling, ECG, etc.) normally scheduled before a surgical intervention.

(see also: Alessandra Ciccozzi, Barbara Pizzi, Alessandro Vittori, Alba Piroli, Gioele Marrocco, Federica Della Vecchia, Marco Cascella, Emiliano Petrucci and Franco MarinangeliThe Perioperative Anesthetic Management of the Pediatric Patient with Special Needs: An Overview of Literature. Children 2022, 9, 1438.)

Comments 3: Discussion section needs good improvement. There is a high lack of literature studies report either comparison.

Response 3: We absolutely agree with this observation. We have expanded the discussion (red part – page 7-8 – lines 231-265). The lack of comparative studies is due to the few studies published on this topic. This prompted us to write this retrospective work, considering that we have been working for many years on adult and pediatric patients with disabilities. Our hope is, in fact, to increase attention on the anesthesiological management of patients with special needs.

Comments 4: Conclusion is vague.

Response 4: Thanks for the suggestion. We have expanded the conclusions (page number 10- lines 347-352)

Comments 5: References section needs to be revised

Response 5: Thanks for the observation. The references have been revised.

Reviewer 4 Report (New Reviewer)

Comments and Suggestions for Authors

              The article had two objectives, the main one was to evaluate the incidence of difficult airway management in special needs pediatric patients that undergo dental treatment under general anesthesia. The second one was to highlight the influence of general anesthesia on the patient outcome, looking at perioperative complications, and evaluating awakening time and hospitalization duration. All the data was collected from the medical reports of the patients admitted to a single medical center in a period of 1 year and 8 months, resulting in 41 eligible cases that were included in the study. There was plenty of data collected and presented in detail in the results section and the conclusions drawn are in concordance with the findings. All patients were ventilated with a face mask, and only 4 (9.75%) out of the 41 patients presented intubation difficulties. Also there were no intra- or postoperative complications attributable to anesthetic conduct.

              The purpose of the study is well explained and the paper could serve as a tool to predict de difficulty of the intubation in special needs patients, based on their disorder. Although the number of medical records analyzed was limited, there was a lot of valuable data extracted. The materials and methods are well described making the results easily reproducible.

Major issues.

1. In Table 1, page 5, Epilepsy is listed as the main disability as well as a comorbidity.

If the main pathology in a patient is epilepsy then the main particularity in the management of the treatment for this individuals is represented by the possible pharmaceutical interactions. With this in mind I don't think they should be included in the "special needs" group as it is defined in the introduction of this article.

2. The relevance of hospitalization duration

I don't think the hospitalization duration is relevant to the purpose of this study considering that it depends more on the type of procedure effectuated and the postoperative care needed by the patient.

(for example: the patients undergoing caries treatment, will naturally have a shorter hospitalization period than the ones with excision of a facial or tongue neoformation )

3. Exclusion criteria  (page 4, lines 179-181)

Thirteen cases were excluded because they referred to dental procedures performed on patients previously treated during the reporting period.

In my opinion this shouldn’t be a reason to not include those cases. Even though it was not the first time the patients underwent dental procedures in general anesthesia, they were still individuals that could be included in the “special needs” group, based on the definition.  

4. There is no information about the minor complications of anesthesia.

Before Discussions, page 8, lines 258-259, there is written that there were no major perioperative complications referable to anesthesia. Also, at page 10, lines 335-336, it is specified that in no case were there any intra- or postoperative complications attributable to anesthetic conduct. (without a mention if the sentence refers to major complications or complications in general).

Minor issues.

1. The first paragraph under Materials and Methods

The whole paragraph is a sentence that is pretty hard to follow. I would suggest to split it in two:

For the execution of the retrospective, observational, single-center study, were examined the records of uncooperative patients admitted to the Department 114 of Maxillofacial Surgery at P.O. San Salvatore in L'Aquila during the period January 2021 115 to August 2022. All the collected data was used with the specific authorization of the Presidio Medical Directorate of the S. Salvatore Civil Hospital in L'Aquila and after obtaining approval to carry out the study from the Ethics Committee for the Provinces of L'Aquila and Teramo (minute No. 25 of September 14, 2022).

2. Table 3 a bit too crowded (page 6, line 203)

If “Oral cavity check/ clearance” was performed in every case maybe the column should contain only the “ Main procedure”.

3. The term “other” (page 5, table 2, line 197)(page 6, table 3, line 203)(page 6, table 5, line 224)

In the mentioned tables is used the term other without defining anywhere what it refers to.

Author Response

Thank you very much for taking the time to review this manuscript. We greatly appreciate the thoroughness of your comments. Please find the detailed responses below and the corresponding revisions/corrections highlighted (in red) changes in the re-submitted files.

Major issues.

Comments 1. In Table 1, page 5, Epilepsy is listed as the main disability as well as a comorbidity.

If the main pathology in a patient is epilepsy then the main particularity in the management of the treatment for this individuals is represented by the possible pharmaceutical interactions. With this in mind I don't think they should be included in the "special needs" group as it is defined in the introduction of this article.

Response 1: Thank you for your observation. Actually, all patients included in the study were “special needs” as reported in the medical records. We reported the pathologies as “main” or “comorbidity” based on what was recorded in the anesthesiology record. Including epilepsy. Patients with only epilepsy (without “special needs”) were not included.

Comments 2: The relevance of hospitalization duration

I don't think the hospitalization duration is relevant to the purpose of this study considering that it depends more on the type of procedure effectuated and the postoperative care needed by the patient.

(for example: the patients undergoing caries treatment, will naturally have a shorter hospitalization period than the ones with excision of a facial or tongue neoformation)

Response 2: Thank you for your observation. We also wanted to collect this data to verify whether the execution of general anesthesia had led to longer hospitalization times (for example: need to spend a night in hospital). This is because for patients with special needs and their families the hospitalization time is a significant stress factor. The data collected showed that almost all patients were discharged the same day, after a few hours from anesthesia. In only one case did the hospitalization last several days (for surgical problems).

 Comments 3: Exclusion criteria (page 4, lines 179-181)

Thirteen cases were excluded because they referred to dental procedures performed on patients previously treated during the reporting period.

In my opinion this shouldn’t be a reason to not include those cases. Even though it was not the first time the patients underwent dental procedures in general anesthesia, they were still individuals that could be included in the “special needs” group, based on the definition.

Response 3: Thank you for your observation. In fact, we discussed this exclusion criteria a lot (also because it excluded 13 cases). The fact of having subjected the same patient to anesthesia several times (even 3 times in a short time) seemed to the majority of the experimenters to be a factor that could alter the results of the study. Therefore, we considered each patient only 1 time (the first).

Comments 4:  There is no information about the minor complications of anesthesia.

Before Discussions, page 8, lines 258-259, there is written that there were no major perioperative complications referable to anesthesia. Also, at page 10, lines 335-336, it is specified that in no case were there any intra- or postoperative complications attributable to anesthetic conduct. (without a mention if the sentence refers to major complications or complications in general).

Response 4: Thank you for pointing this out. We have corrected: “In no case were there any intra- or postoperative complications, minor or major, attributable to anesthetic conduct,” (page 10, lines 335)

Minor issues.

Comments 1: The first paragraph under Materials and Methods

The whole paragraph is a sentence that is pretty hard to follow. I would suggest to split it in two:

For the execution of the retrospective, observational, single-center study, were examined the records of uncooperative patients admitted to the Department 114 of Maxillofacial Surgery at P.O. San Salvatore in L'Aquila during the period January 2021 115 to August 2022. All the collected data was used with the specific authorization of the Presidio Medical Directorate of the S. Salvatore Civil Hospital in L'Aquila and after obtaining approval to carry out the study from the Ethics Committee for the Provinces of L'Aquila and Teramo (minute No. 25 of September 14, 2022).

Response 1: Thanks for the suggestion. The change has been made (page 3; lines 110-116)

Comments 2: Table 3 a bit too crowded (page 6, line 203)

If “Oral cavity check/ clearance” was performed in every case maybe the column should contain only the “Main procedure”.

Response 2: Thanks for the suggestion. Table 3 has been modified (page 6; line 203)

Comments 3. The term “other” (page 5, table 2, line 197) (page 6, table 3, line 203) (page 6, table 5, line 224)

In the mentioned tables is used the term other without defining anywhere what it refers to.

Response 3: Thanks for suggestion. We have inserted the missing information in Table 3 (page 5, line 197) and table 5 (page 6, line 224). As for table 2 the term “other” indicates drugs taken by only 1 patient, without interest for anesthesia. We preferred not to list them to avoid making the table too crowded. We have, however, added the note “*drug taken by one patient only” (pag 5, line 197)

Round 2

Reviewer 1 Report (Previous Reviewer 3)

Comments and Suggestions for Authors

previous comments stand

Reviewer 3 Report (New Reviewer)

Comments and Suggestions for Authors

The revised manuscript is improved substantially.

Comments on the Quality of English Language

The manuscript should be double checked for spelling and grammar.

This manuscript is a resubmission of an earlier submission. The following is a list of the peer review reports and author responses from that submission.

Round 1

Reviewer 1 Report

Comments and Suggestions for Authors

The Sample size is quite small.

The supporting statistical analysies is poorly represented and insufficiently exposed.

Comments on the Quality of English Language

I don't have any comment

Author Response

Thank you very much for taking the time to review this manuscript. Please find the detailed responses below and the corresponding revisions/corrections highlighted (in red) changes in the re-submitted files.

Comments 1: The Sample size is quite small.

Response 1: Thank you for pointing this out. We agree with this comment. It is certainly a limitation of the study. In fact, we added the subsection “study limitations”: “Another limitation of the study is the small size of the sample, mainly due to the Covid-19 health emergency present in Italy in the period under consideration, which numerically limited the execution of all elective operations, including dental practices.” ( page number 10,  lines 371-381)

Comments 2: The supporting statistical analysies is poorly represented and insufficiently exposed.

Response2: Thank you for your observation, we have expanded the description of the statistical analysis in Materials and Methods: “Statistical analysis was conducted using "The Sas System version 9.4." Descriptive analysis was carried out using the mean, standard deviation and range for continuous quantitative variables and with frequencies for nonquantitative variables. After assessing the normality of the distribution of the variables with the Shapiro-Wilk test, statistical comparisons were carried out using the non-parametric Wilcoxon Rank Sum test.” (page number 4,  lines 169-173)

Reviewer 2 Report

Comments and Suggestions for Authors

1I thank the editor for the opportunity to review this manuscript. Please consider the following recommedations:

.      

Patients between 3-8 yo are not patients with SN, are, simply, pediatric patients, which have a higher level of anxiety

2.       Introduction: It is not very clear stated if your patients cohort has additionally psychosocial impairments or they are just pediatric patients

3.       Methodology: It is obvious that some patients (6%- ASA I) have no disability (including SN), which disagrees the initial objective of your study (lines 64-67)

4.       Table 1- the subsections main disability and comorbidity are confounding. Please reconsider or define them precisely.

5.       Lines 169-172, the four patients included in your study are not suitable from the perspective of the title. Please exclude from your study these patients and reconsider the methodology and the results in the light of the new percentages.

6.       Lines 173-178- please reconsider it a study limitations

7.       Lines 236-280- is more suitable for the introduction subsection. Please reconsider.   

8.       Please consider a study limitations subsection

Author Response

Thank you very much for taking the time to review this manuscript. Please find the detailed responses below and the corresponding revisions/corrections highlighted (in red) changes in the re-submitted files.

Comments 1: Patients between 3-8 yo are not patients with SN, are, simply, pediatric patients, which have a higher level of anxiety:

Response 1: Thank you for pointing this out. We agree with this comment, however we considered pediatric patients with special needs based on what is reported in the literature: International scientific literature defines pediatric patients with “special needs” (SN) as children suffering from psycho-physical disorders with related relational and cognitive problems.

-Caicedo, C. Families with Special Needs Children: Family Health, Functioning, and Care Burden. J. Am. Psychiatr. Nurses Assoc. 2014, 20, 398–407.

-Huang, L.; Freed, G.L.; Dalziel, K. ChildrenWith Special Health Care Needs: How Special Are Their Health Care Needs? Acad. Pediatr. 2020, 20, 1109–1115.

-Alessandra Ciccozzi, Barbara Pizzi, Alessandro Vittori, Alba Piroli, Gioele Marrocco, Federica Della Vecchia, Marco Cascella, Emiliano Petrucci and Franco Marinangeli. The Perioperative Anesthetic Management of the Pediatric Patient with Special Needs: An Overview of Literature. Children 2022, 9, 1438.

Comments 2: Introduction: It is not very clear stated if your patients cohort has additionally psychosocial impairments or they are just pediatric patients:

Response 2: Thanks for the observation. We have added: “with psychophysical and/or cognitive disorders” (page number 3, lines 102, 103)

Comments 3:  Methodology: It is obvious that some patients (6%- ASA I) have no disability (including SN), which disagrees the initial objective of your study (lines 64-67)

Response 3: Thank you for pointing this out. In reality, the inclusion criteria in the study included: “all uncooperative patients aged between 3 and 17 years old, with ASA status I-III, undergoing dental surgery under general anesthesia during the period under consideration were included in the study.” (page number 3, lines 122-123).

Indeed, 6 pediatric patients, at the time of the anesthetic visit, were classified as ASA Status I. These were children with relational disorders (therefore included in the definition of patients with Special Needs), but physically healthy, without comorbidities (therefore classified ASA STATUS I). This is why they were included in the study.

Comments 4: Table 1- the subsections main disability and comorbidity are confounding. Please reconsider or define them precisely.

Response 4: Thank you for pointing that out. We have better defined in the text the main disabilities and comorbidities: "…including the main disability (disorder reported in the medical record as the main one) and the comorbidities presented (disorders coexisting with the main pathology). (page number 4, lines 186,187)

Comments 5: Lines 169-172, the four patients included in your study are not suitable from the perspective of the title. Please exclude from your study these patients and reconsider the methodology and the results in the light of the new percentages.

Response 5: Thank you for your observation. However, we would like to point out that the 4 patients also underwent dental surgery under general anesthesia (and therefore included in the study). In addition, taking advantage of the general anesthesia performed for the dental procedure, other medical procedures were also performed. This possibility is very important for patients with SN: being able to perform medical visits or procedures taking advantage of general anesthesia.

In the text we clarified the concept better: “Furthermore, in 4 patients, in addition to the planned dental surgery, other medical procedures were performed, taking advantage of the general anesthesia performed: a urological examination with replacement of the suprapubic catheter, a gastroenterological examination with esophagogastroduodenoscopy (EGDS), an ENT examination and a percutaneous endoscopic gastrostomy (PEG) replacement”. (page number 6, lines 208-212)

Comments 6: Lines 173-178- please reconsider it a study limitations

Response 6: Please, see explanation in point 5

Comments 7:  Lines 236-280- is more suitable for the introduction subsection. Please reconsider.   

Response 7: Thank you for pointing this out. We agree with this comment. Therefore, we have moved the indicated part from the Discussion subsection to the Introduction subsection (page number 2, lines 61-96), as suggested. We have, consequently, modified the references.

Comments 8: Please consider a study limitations subsection

Response 8: Thanks for the observation. We agree with this comment. Therefore, we added the subsection “study limitations”, as suggested (page number 10, lines 371-381)

Reviewer 3 Report

Comments and Suggestions for Authors

This is an audit of syndrome c/ unco operative paediatric patients presenting for dental examination Under GA. Unfortunately, this audit does not add anything new to teh existing knowledge and offers no insight as to how these cases should be managed. Another shortcoming is that ten dataset is too small and restricted to a single centre. 

Author Response

Thank you very much for taking the time to review this manuscript. Please find the detailed responses below and the corresponding revisions/corrections highlighted (in red) changes in the re-submitted files.

Comments 1: This is an audit of syndrome c/ unco operative paediatric patients presenting for dental examination Under GA. Unfortunately, this audit does not add anything new to teh existing knowledge and offers no insight as to how these cases should be managed. Another shortcoming is that ten dataset is too small and restricted to a single centre.

Response 1: Thank you for pointing this out. We agree with this comment. In our region, however, few medical centers accept to treat patients with disabilities, especially pediatric ones, to perform dental treatments under general anesthesia, precisely because of the complexity of perioperative management and the risks associated with performing anesthesia. We hope that even a small single-center study reporting its favorable experience can help other healthcare professionals to better address the problem.

Due to the small sample size, we added the subsession “Study limitations”. (page number 10, lines 371-381)